

# Delivering a medical school elective with massive open online course (MOOC) technology

Robert Robinson

Internal Medicine, Southern Illinois University School of Medicine, Springfield, IL, USA

Corresponding author
Robert Robinson,
rrobinson@siumed.edu

## ABSTRACT

**Introduction:** The educational technology of massive open online courses (MOOCs) has been successfully applied in a wide variety of disciplines and are an intense focus of educational research at this time. Educators are now looking to MOOC technology as a means to improve professional medical education, but very little is known about how medical MOOCs compare with traditional content delivery.

**Methods:** A retrospective analysis of the course evaluations for the Medicine as a Business elective by fourth-year medical students at Southern Illinois University School of Medicine (SIU-SOM) for the 2012–2015 academic years was conducted. This course was delivered by small group flipped classroom discussions for 2012–2014 and delivered via MOOC technology in 2015. Learner ratings were compared between the two course delivery methods using routinely collected course evaluations.

**Results:** Course enrollment has ranged from 6–19 students per year in the 2012–2015 academic years. Student evaluations of the course are favorable in the areas of effective teaching, accurate course objectives, meeting personal learning objectives, recommending the course to other students, and overall when rated on a 5-point Likert scale. The majority of all student ratings (76–95%) of this elective course are for the highest possible choice (Strongly agree or Excellent) for any criteria, regardless if the course was delivered via a traditional or MOOC format. Statistical analysis of these ratings suggests that the Effective Teacher and Overall Evaluations did not statistically differ between the two delivery formats.

**Discussion:** Student ratings of this elective course were highly similar when delivered in a flipped classroom format or by using MOOC technology. The primary advantage of this new course format is flexibility of time and place for learners, allowing them to complete the course objectives when convenient for them. The course evaluations suggest this is a change that is acceptable to the target audience.

**Conclusions:** This study suggests that learner evaluations of a fourth-year medical school elective course do not significantly differ when delivered by flipped classroom group discussions or via MOOC technology in a very small single center observational study. Further investigation is required to determine if this delivery method is an acceptable and effective means of teaching in the medical school environment.

## INTRODUCTION

The educational technology of massive open online courses (MOOCs) has been successfully applied in a wide variety of disciplines and are an intense focus of educational research at this time (*Bozkurt et al., 2015*). MOOCs are a disruptive force in education because they challenge the tradition of lectures and decentralize the education experience in a learner centered way. Learners have embraced this approach, giving some MOOCs an enrollment of over 100,000 (*Mehta et al., 2013*). Educators are now looking to MOOC technology as a means to improve professional medical education (*Harder, 2013*; *Mehta et al., 2013*).

Hundreds of medical MOOCs exist for topics ranging from the Ebola virus to medical informatics (*Grobusch & Browne, 2015*; *Paton, 2014*; *Liyanagunawardena & Williams, 2014*), but very little is known about how these medical MOOCs compare with traditional content delivery.

MOOC delivery platforms allow educators to create and deliver interactive courses with videos, online resources, quizzes, virtual patients (*Stathakarou, Zary & Kononowicz, 2014*; *Kononowicz et al., 2015*) and an ability to interact with other students taking the course. This technology could allow students to learn at a time and place of their choosing, freeing valuable curricular time for hands on experiences. Despite these potential advantages, medical schools have been slow to explore MOOC technology for content delivery.

This study aims to compare learner evaluations and ratings of a course that was previously delivered by traditional methods (in person lecture and case discussions) that is now delivered as a MOOC. The hypothesis is that learner ratings of the course will not significantly differ between the new format and the previous format. The results of this investigation could have significant implications for how medical education is delivered.

## MATERIALS AND METHODS

Course evaluations for the Medicine as a Business elective (Course IM 45434) by fourth-year medical students at Southern Illinois University School of Medicine (SIU-SOM) were collected as customary for the 2012–2015 academic years. The SIU-SOM is located in Springfield, Illinois.

The Medicine as a Business course was offered for the first time in the 2012 academic year and used small group flipped classroom format as an extended nonclinical elective at SIU-SOM. Extended electives are non-overlapping and are scheduled on Thursday afternoons in five week blocks at SIU-SOM. The Medicine as a Business course could start every five weeks during the academic year if two or more students wished to enroll in a specific five week block. Attendance and participation in all discussions was required for successful course completion.

This course was converted to an on demand MOOC format hosted at http://www.Udemy.com for the 2015 academic year to meet student requests for greater flexibility in course content delivery. Course content is delivered in the form of video presentations with associated reading materials and multiple choice questions that could be accessed on a smartphone, tablet, or traditional computer via the internet when convenient for the student. Completion of all course sections was required for successful completion

of the course. Each student started and completed the course independently and had the option of meeting with the faculty to discuss course content. The MOOC content delivery system included a system to send messages to the course instructor and a discussion forum accessible by learners.

The objectives for Medicine as a Business are:

- Understand and apply medical documentation rules
- Understand the medical billing process
- Understand medical coding terminology and resources
- Understand the medical practice revenue cycle
- Evaluate physician productivity using a variety of measures

The course is available online at http://www.udemy.com/business-of-medicine. Course registration is free and open to anyone.

SIU-SOM compiles de-identified aggregate course evaluations for elective faculty for use in course improvement. These course evaluations were compared before and after transition to the MOOC format and do not include any demographic data for the students enrolled. This data was compared for differences between traditional and MOOC based content delivery.

The SIU-SOM elective course evaluation includes:

- Was the faculty an effective teacher?
- Stated course objectives accurately reflected the course.
- I was able to meet my personal learning objectives.
- I would recommend this elective to other students.
- Overall rating of the course.

These items are rated on a 5-point Likert scales.

The rating scale for the overall course rating was 1 = Poor, 2 = Below average, 3 = Average, 4 = Above average, 5 = Excellent.

The rating scale for all other measures was 1 = Strongly disagree, 2 = Disagree, 3 = Neither agree or disagree, 4 = Agree, 5 = Strongly agree.

Differences between items in the course evaluation questions will be compared with the t-test to determine if any significant differences exist.

Statistical analyses were performed using SPSS version 22 (SPSS Inc., Chicago, IL, USA). Two sided p-values < 0.05 were considered significant.

Cohen's d was calculated for each result using Cohen's d calculator for unequal sample sizes developed by *Stangroom (2016)*.

Post-hoc power analysis was conducted with a Post-hoc Statistical Power Calculator for a Student t-test developed by *Soper (2016)*.

Institutional review board review for this study was obtained from the Springfield Committee for Research Involving Human Subjects. This study was determined to not meet criteria for research involving human subjects according to 45 CFR 46.101 and 45 CFR 46.102.

Table 1 **Fourth-year medical student demographics by year.**

| Year | Students | Women | Students enrolled in Medicine as a Business elective |
|------|----------|-------|------------------------------------------------------|
| 2012 | 72 | 34 (47%) | 8 (11%) |
| 2013 | 75 | 35 (47%) | 7 (9%) |
| 2014 | 69 | 31 (45%) | 6 (9%) |
| 2015 | 70 | 38 (54%) | 19 (27%) |

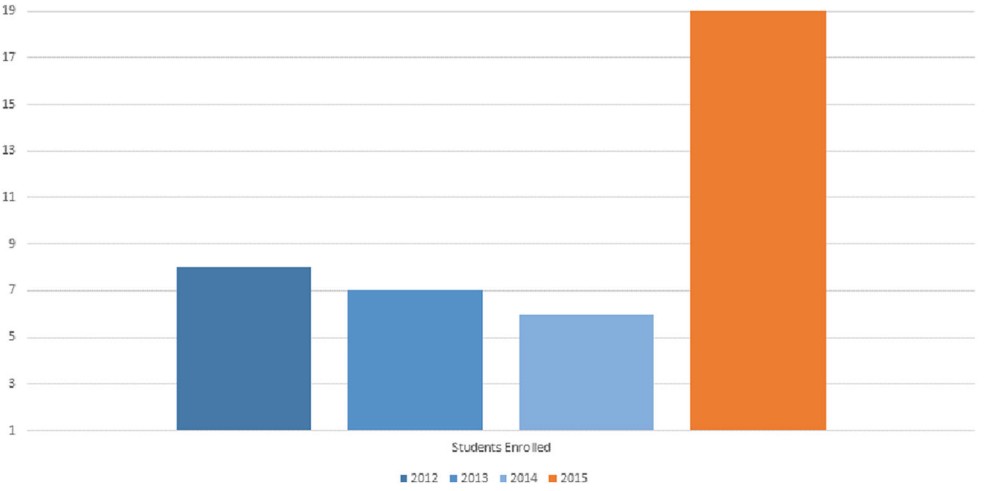

**Figure 1 Student enrollment in the Medicine as a Business elective by academic year.**

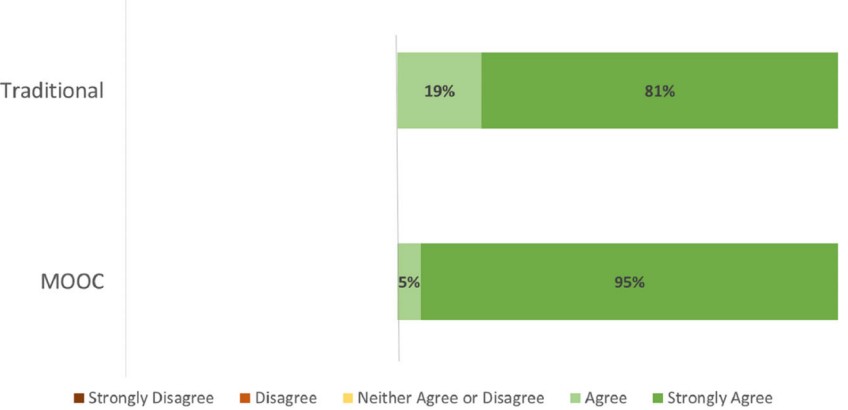

**Figure 2 Student ratings for "Was the course faculty an effective teacher?" by course format.**

## RESULTS

The Business of Medicine elective has been offered for four years at SIU-SOM starting in the 2012 academic year. The size of the fourth-year class at SIU-SOM ranged from 69–75 students during the study period. Women made up 45–54% of the fourth-year class during that timeframe (Table 1). Course enrollment has ranged from 6–19 students per year as shown in Fig. 1 and Table 1.

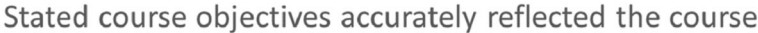

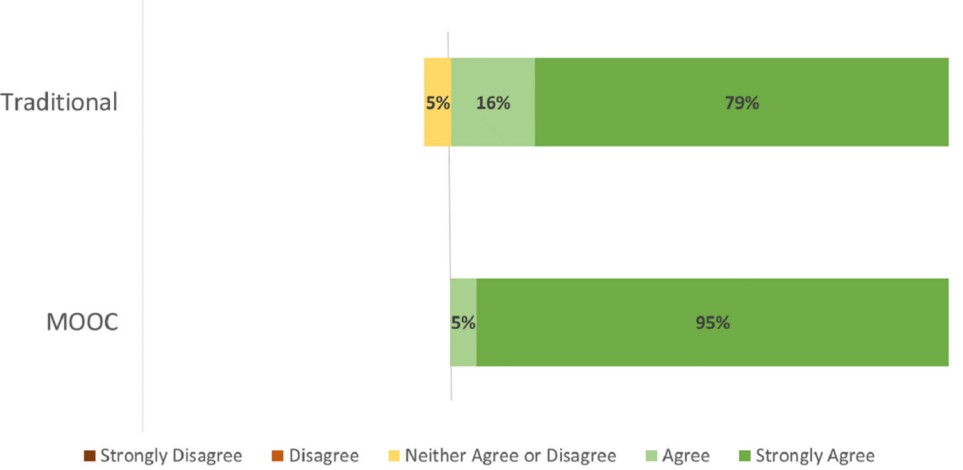

**Figure 3 Student ratings for "Stated course objectives accurately reflected the course" by course format.**

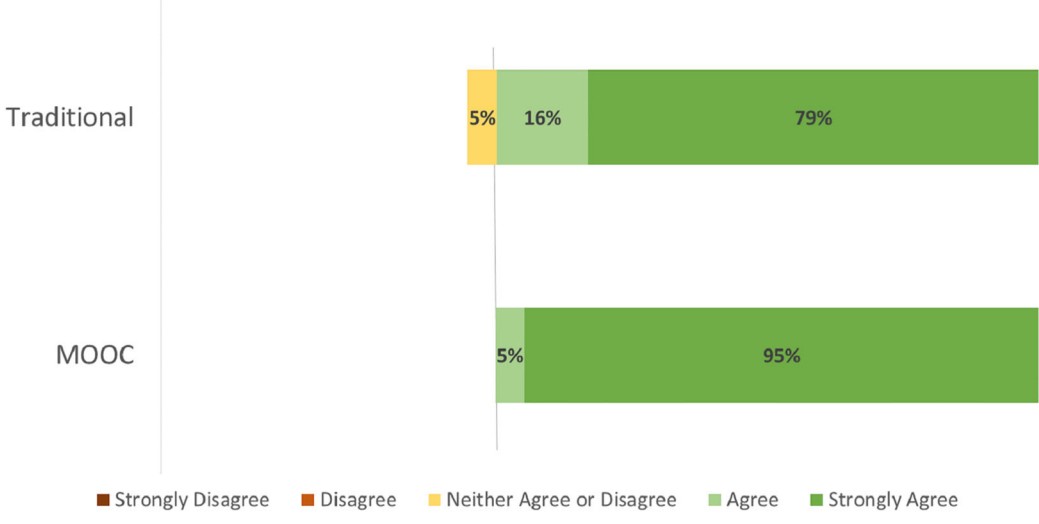

**Figure 4 Student ratings "I was able to meet my personal learning objectives" by course format.**

Student evaluation distributions by course format (MOOC vs. Traditional) for the evaluation criteria of Effective Teacher (Fig. 2), Course Objectives Accurate (Fig. 3), Met Personal Objectives (Fig. 4), Recommend Course (Fig. 5), and Overall Evaluation (Fig. 6) are skewed towards the highest rating and are very similar. No negative course ratings (Disagree, Strongly disagree, Poor, or Below average) were reported for any course evaluation. The majority of all ratings (76–95%) are for the highest possible choice (Strongly agree or Excellent).
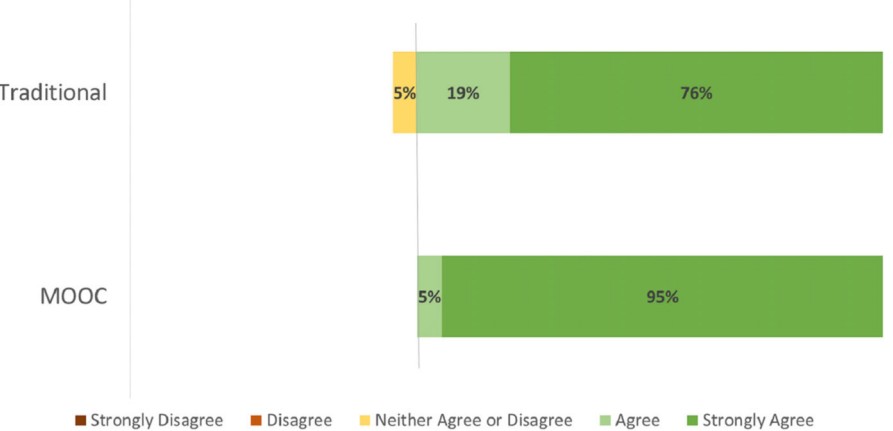

Figure 5 Student ratings "I would recommend this elective to other students" by course format.

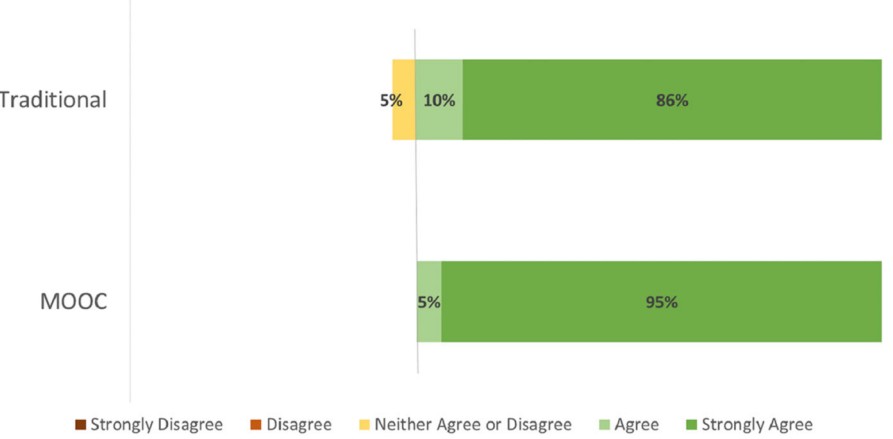

Figure 6 Student ratings for Overall Evaluation by course format.

Table 2 Student course ratings by course format (MOOC vs. Traditional).

| Rating | Traditional mean (SD) N = 21 | MOOC mean (SD) N = 19 | Significance | Power | Cohen's d |
|---|---|---|---|---|---|
| Overall evaluation | 4.81 (0.512) | 4.5 (0.229) | 0.287 | 92% | 0.782 |
| I would recommend this elective to other students | 4.71 (0.561) | 4.5 (0.229) | 0.100 | 45% | 0.490 |
| I was able to meet my personal learning objectives | 4.75 (0.550) | 4.5 (0.229) | 0.156 | 67% | 0.593 |
| Stated course objectives accurately reflected the course | 4.70 (0.571) | 4.5 (0.229) | 0.087 | 39% | 0.460 |
| Was the course faculty an effective teacher? | 4.90 (0.301) | 4.5 (0.229) | 0.620 | 99% | 1.496 |

Student evaluations by course format (MOOC vs. Traditional) show no statistically significant differences (p = NS for all criteria, Table 2). A medium or greater effect sizes were seen for the Effective Teacher (Cohen's d = 0.782), Met Personal Objectives (Cohen's d = 0.593), and Overall Evaluation (Cohen's d = 1.496) criteria. Small effect sizes were seen for the Course Objectives Accurate (Cohen's d = 0.460) and Recommend Course to other students (Cohen's d = 0.490) criteria. Post-hoc power analysis indicated the results for Effective Teacher (99%) and Overall Evaluation (92%) had greater than 80% power to correctly reject the null hypothesis. The results for all other criteria did not have sufficient power to correctly reject the null hypothesis.

## DISCUSSION

The majority of all student ratings (76–95%) of this elective course are for the highest possible choice (Strongly agree or Excellent) for any criteria, regardless if the course was delivered via a traditional or MOOC format. Statistical analysis of these ratings suggests that the Effective Teacher and Overall Evaluations no not statistically differ between the two delivery formats.

The primary advantage of this new course format is flexibility of time and place for learners, allowing learners to take the course when and where convenient for them instead of in five week blocks on campus. The course evaluations suggest this change is acceptable to the target audience.

Enrollment in this nonclinical elective was good, ranging from 9–27% of the students each year. It is unclear why course enrollment was highest during the 2015 academic year (27% of all fourth-year students taking the elective). This increase in enrollment may also be accompanied by other demographic changes in the students taking this elective course that may influence course ratings. However, specific student demographics (gender, race, residency choice, and other factors) are not available for analysis to protect the anonymity of students enrolled in this course.

The rating system used for electives at SIU-SOM are not ideal to evaluate a course delivered in this manner. However, the highly similar Effective Teacher and Overall course ratings are likely to represent an acceptance of the course format change.

The course subject, the business aspects of a medical practice, is likely to be more amenable to delivery via MOOC technology than many other subject areas in medical education. However, the University of California-San Francisco (UCSF) has successfully offered a clinical problem solving MOOC with tens of thousands of learners, suggesting that complex clinical education can be delivered by this format (*Harder, 2013*).

The numbers of students enrolled in this course before and after the change to the MOOC format are very small, increasing the risk of a Type 2 error due to insufficient statistical power. Post hoc power analysis showed a range of power from 39 to 99% for the evaluation criteria. Only two measures, Effective Teacher and the Overall Evaluation, had sufficient power to correctly reject the null hypothesis. Challenges with statistical power reflect the small sample size in this study and a course rating distribution strongly skewed towards the highest rating for all areas evaluated.

This study is a single institution observational study with limited follow up. These limitations may reduce the generalizability of the results of this study.

## CONCLUSIONS

This study suggests that learner evaluations of a fourth-year medical school elective course may not significantly differ when delivered in flipped classroom discussion or via MOOC technology in a very small single center observational study.

Further investigation is required to determine if this delivery method is an acceptable and effective means of teaching in the medical school environment.

### Funding
The author received no funding for this work.

### Competing Interests
The author declares that they have no competing interests.

### Author Contributions
- Robert Robinson conceived and designed the experiments, performed the experiments, analyzed the data, contributed reagents/materials/analysis tools, wrote the paper, prepared figures and/or tables, reviewed drafts of the paper.

### Human Ethics
The following information was supplied relating to ethical approvals (i.e., approving body and any reference numbers):

Institutional review board review for this study was obtained from the Springfield Committee for Research Involving Human Subjects. This study was determined to not meet criteria for research involving human subjects according to 45 CFR 46.101 and 45 CFR 46.102.

### Data Deposition
The raw data has been supplied as Supplemental Dataset Files.

### Supplemental Information
Supplemental information for this article can be found online at http://dx.doi.org/10.7717/peerj.2343#supplemental-information.

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
