# Peer review of "Delivering a medical school elective with massive open online course (MOOC) technology"

_PeerJ, doi:10.7717/peerj.2343_

## Round 0.1 · original submission · Minor Revisions

Please describe the demographics of students enrolled in each course.
Please address how the data might be impacted by potential differences in the enrollment and demographics between the two data sets.

Please include additional description of the assessments, discussion format, pace, attendance/online access in traditional versus MOOC courses.

Please address reviewer 1 comments about sample size and statistical power by including a discussion about your sample size limitations. Reviewers 2 and 3 make good comments that should also be addressed.

·

Basic reporting

Basic reporting is fine. The paper is clear and well written.

Experimental design

The major flaw in this paper is the lack of sample size estimation and consequent power limitations. The experimental design is also not strong - since numbers are different between the year in with the MOOC was used and previous years, this suggests different types of students took the elective in different years and there is no description of the students enroling. However, I do appreciate the difficulty of randomising to different groups.

Validity of the findings

The numbers are very small and lack of a statistically significant difference is likely a Type 2 error due to inadequate power to demonstrate a difference. There is no discussion of why the numbers were so much larger in the year that the MOOC was the method of instruction. There were no process measures – how many accessed each part of the MOOC?

Additional comments

This is an important area of research, and I congratulate you on attempting it. However, you need to pay more attention to the research design, in particular the power requirements to demonstrate a difference.

Reviewer 2 ·

Basic reporting

The manuscript addresses a narrow but important topic - to what extent, in relation to achievement of learning objectives, do MOOC technology facilitate achievement of learning objectives compared with traditional teaching in medical education. The author focused on a non-clinical course that was previously taught in traditional on-campus format for several years and compared learning outcomes with a MOOCs format. The basis reporting aspects are well presented - pass.

Experimental design

The author examined the student evaluations of four different cohorts for the sam course. It is not stated if the MOOCs format is completely self-paced, but I presume it is. It is difficult to measure the "Effective Teacher" component if there is no regular (or on-demand) interaction between the learners and the course facilitator. Also, the "course objectives accurate" evaluation criterion is vague. I expect the learners to evaluate if the course objectives were met, not whether or not they were accurate " conditional fail, subject to clarification".

The experimental design might have been improved by relating aspects of the MOOCs course to specific learning objectives and requiring learners to evaluate the extent to which specific topics met corresponding learning objective. The article fails in this regard

Validity of the findings

The findings are valid, but the implications of the study are constrained by the inadequately explained experimental design. - "pass"

Additional comments

No comments

·

Basic reporting

No Comments

Experimental design

The experimental design seems fine. Ideally, you would have larger sample sizes and more than one year worth of data after making the change to MOOC, but it is workable as written.

Validity of the findings

The statistical methods applied and the data as presented seem accurate. I do have a couple of suggestions: at line 90, you note that "the primary advantage of this new course format is flexibility of time and place". Did you survey course participants about what they felt were the primary advantages to the format change? If not, you may want to reword to clarify that. You may also want to note some of the other advantages as well, particularly for the institution. Enrollment in this course more than doubled: perhaps there were students who wanted to take the course as a live class but couldn't.

Additionally, at line 45, you note that the course content of the MOOC is video and reading material based. Given that the old format of the course was group discussion, it seems like a fairly major change in format. Can you confirm that class members no longer have discussions related to the material? Is there an online forum-based discussion component? That's fairly common with MOOCs. If not, it's worth mentioning the shift in format.

Overall, a nice job of presenting the basics of MOOcs, and the data from your survey.

Additional comments

A nicely written article using a limited sample size.

---

## Round 0.2 · Minor Revisions

As reviewer 1 mentions, the limitations of the study need to be mentioned in the abstract.

Thank you for adding a sentence in your manuscript discussion about the possibility of type 2 error. But I am afraid that this might not be enough to completely disclose the quality of your data. I agree with reviewer 1 in that you need to provide additional statistical parameters. For example, did you perform a power analysis? I understand that you are limited by the number of students enrolled, but this information needs to be disclosed. Also, there is a lot of discussion regarding effect size and whether or not the effect size needs to be reported for non-significant data. But I would say this: in this type of analysis it is impossible to distinguish a null effect from a very small effect, so in this case one needs to consider that a high p-value is not evidence that the null hypothesis is true (type 2 error). Could you provide effect sizes (and maybe confidence intervals?) to strengthen your manuscript? Alternatively, include statistic power information and write an extensive discussion of why other parameters are not included (and what it means for your conclusions). In the case you do not provide sample size and CI, please rewrite your article so your wording doesn't imply that you are accepting the null hypothesis. Please mention this information in the abstract as well. One example: the abstract ends with the sentence: MOOCs may be a reasonable format to deliver medical school courses. Please remove or rewrite limiting your conclusions to the data you have: non-significant self-reported student's evaluations. Reasonable in what sense? what exactly is your null hypothesis?
I am aware of table 1 describing the demographics of the 4th year class. But since the enrollment to the courses you describe are only a small percentage of that class it doesn't reflect potential differences in the courses you are trying to compare. Thank you for clarifying that demographics on a course by course basis are not provided to faculty. You will need to be a lot more clear in your text about this then.

·

Basic reporting

Basic reporting is fine, although the abstract does not fully reflect the content of the paper. In particular, the limitations of the study are not included in the abstract which has not been changed since the original submission.

Experimental design

Very weak, although some of the limmitations are now included in the paper.

Validity of the findings

Very weak. The possibility of a Type two error are mentioned, but no further statistical information given.

Additional comments

Thanks for making some changes to the text, which have improved the paper, but the fundamental weaknesses remain.

·

Basic reporting

The updates provided have addressed any issues I had with the article.

Experimental design

The updates provided have addressed any issues I had with the article.

Validity of the findings

The updates provided have addressed any issues I had with the article.

---

## Round 0.3 · accepted · Accept

Thanks for addressing the changes we requested!